# MVPbev: Multi-view Perspective Image Generation from BEV with Test-time Controllability and Generalizability

## ABSTRACT

This work aims to address the multi-view perspective RGB generation from text prompts given Bird-Eye-View(BEV) semantics. Unlike prior methods that neglect layout consistency, lack the ability to handle detailed text prompts, or are incapable of generalizing to unseen view points, MVPbev simultaneously generates cross-view consistent images of different perspective views with a two-stage design, allowing object-level control and novel view generation at test-time. Specifically, MVPbev firstly projects given BEV semantics to perspective view with camera parameters, empowering the model to generalize to unseen view points. Then we introduce a multi-view attention module where special initialization and denoising processes are introduced to explicitly enforce local consistency among overlapping views w.r.t. cross-view homography. Last but not the least, MVPbev further allows test-time instance-level controllability by refining a pre-trained text-to-image diffusion model. Our extensive experiments on NuScenes demonstrate that our method is capable of generating high-resolution photorealistic images from text descriptions with thousands of training samples, surpassing the state-of-the-art methods under various evaluation metrics. We further demonstrate the advances of our method in terms of generalizability and controllability with the help of novel evaluation metrics and comprehensive human analysis. Our code and model will be made available.

## CCS CONCEPTS

• **Computing methodologies** → *Scene understanding*; **Computer vision tasks**.

## KEYWORDS

Test-time controllability, cross-view consistency, image generation

## 1 INTRODUCTION

Multi-view perspective images are beneficial for autonomous driving tasks [2]. Nowadays, multi-view cameras, including ones mounted in the front and on the side, have become basic requirements in large driving datasets, such as NuScenes [2], Argoverse [4] and Waymo [26]. Typically, images from multiple cameras' views are perceived and further represented in Bird-Eye-View(BEV) [32], where downstream tasks such as prediction and planning take place later on [7, 19]. Intuitively, the BEV allows more interpretability as it

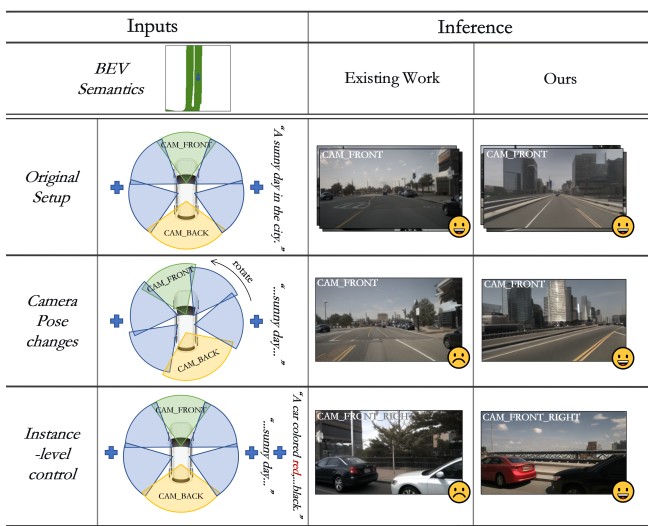

**Figure 1: Our MVPbev is able to generate multi-view perspective images with BEV semantics and text prompts. More importantly, MVPbev allows test-time view-point and instance-level control, significantly improving its generalizability.**

provides a tangible interface to the real world, thus is beneficial and practical for higher-level modeling and decision making [16, 17].

Though being of great importance in autonomous driving tasks, reliable BEV representation requires a large amount of data during the training stage, which can be time-consuming to obtain or annotate. One intuitive solution to this data issue is to obtain diverse perspective RGB images as well as their corresponding BEV semantics with generative models. Diverse yet plausible BEV semantics, compared to their corresponding perspective RGB or semantics, are much easier to simulate in a realistic manner with the help of parametric representations [32]. To this end, it is natural and practical to assume that BEV semantics, rather than perspective RGB images, are given. Then the remaining question is to generate cross-view visually and semantically consistent photorealistic RGB images with known BEV semantics.

Despite the progress of generative models with given constraints [34], there are three main drawbacks in existing attempts to address this cross-view image generation problem [9, 27, 29]. Firstly, existing frameworks rely heavily on the training samples, leading to unsatisfactory test-time controllability. For instance, changing camera poses or providing extra control on object instance is beyond prior art. Moreover, cross-view consistency is not well enforced, resulting in inconsistent visual effects in overlapping FOVs. Finally, no thorough human analysis is performed on image generation tasks, resulting in un-interpretable comparison results.

To this end, we propose a novel two-stage method MVPbev that aims to generate controllable multi-view perspective RGB images with given BEV semantics and text prompts by explicitly enforcing cross-view consistency (See Fig. 1.a). Unlike existing work that lacks the test-time generalizability, MVPbev further allows both view-point and detailed text prompt changes at test-time, providing satisfactory performances under human analysis without requiring additional training data. To achieve that, MVPbev consists of two stages, or view projection and scene generation stages. The former stage transforms the given BEV semantics to multiple perspective view w.r.t. camera parameters. On the one hand, it enforces global consistency across views with explicit geometric transformation. On the other hand, such a design decouples the two stages, allowing the second stage to better capture view-point-invariant properties. The second stage of MVPbev starts from a pre-trained stable diffusion (SD) model. By explicitly incorporating a cross-view consistent module, together with our noise initialization and de-noising processes design, it can produce multi-view visually consistent and photo-realistic images, especially at overlapping FOVs. To further improve the test-time generalizability on objects, our MVPbev handles foreground instances and background layout individually, leading to better controllability during inference.

We validate our ideas on NuScenes [2] and follow the standard split. In contrast to methods that focus on improvements over downstream tasks or semantic consistency, we include additional extensive human analysis, especially on visual consistency over overlapping FOVs, across multiple views, and test-time view point and text prompt changes. We demonstrate that our proposed method not only provides better test-time controllability and generalizability, but also gives high-quality cross-view RGB images.

In short, our contribution can be summarized as follows:

- A novel multi-view image generation method that is capable of producing semantically and visually consistent perspective RGB images from BEV semantics, with only thousands of images as training data.
- A more controllable yet extendable algorithm that gives realistic perspective RGB images.
- State-of-the-art performances on large driving datasets, with comprehensive human analysis.

## 2 RELATED WORK

Image editing [20] and generation [24] are heated topics in computer vision. Though this can be related to a vast literature, we would focus on two lines of work, conditional image generation and novel view image synthesis, as they are closely relevant.

**Conditional image generation** Generative models, e.g., Gaussian Mixture model [23] and Bayesian network [11], have been a long-term research problem in machine learning and computer vision as it is able to explain complex data distribution. In particular, generative models of images are not only of great importance for unsupervised feature learning, but also enable applications such as image editing [13, 20]. With the rise of deep learning techniques, such as auto-regressive models [3], variational autoencoders (VAEs) [14], and generative adversarial networks (GANs) [10], as well as the emerging of huge amount of data [8], we observe photo-realistic images with very good quality. Among them, conditional

GANs have been well explored where various constraints, including discrete labels, text, and images, are considered. More recently, stable diffusion models [24] are widely used to generate detailed images conditioned on text descriptions. Compared to the prior art, they not only demonstrate SOTA image generation quality, but also showcase great generalizability with the help of foundation models [15]. Later on, Controlnet [34] largely improves the overall performance of diffusion models without losing the original robustness by allowing a diverse set of conditional controls, e.g., depth, semantics, or sketches. Despite impressive progress, multi-view or cross-view text-to-image generation still confronts issues of computational efficiency and consistency across views. To this end, MVDiffusion [29] proposes a novel correspondence-aware attention module to create multi-view images from text with the ability to maintain global correspondence. Though providing good multi-view RGB images, MVDiffusion fails to generalize to more dramatic viewpoint changes or smaller overlapping areas. Perhaps the co-current work, including BEVGen [27], BEVControl [33], and MagicDrive [9], are the closest to ours. The first one generates multi-view visual consistent images based on the BEV semantics by employing an auto-regressive transformer with cross-view attention. While the last two work with image sketches/semantics and text, and utilizes cross-view cross-object attention to focus more on consistency on individual contents. However, none of existing work allows test-time generalizability, e.g., view-point changes or detailed instance-level text prompts. Nor do they conduct human analysis on image generation quality. In contrast, we propose to exploit both global and local consistency to leverage semantic and visual coherency, together with our training-free objects control method to enforce detailed instance-level control. Moreover, we provide comprehensive human analysis to demonstrate the effectiveness of our method in a more reliable manner.

**Novel view image synthesis** There are two broad categories in which the novel view synthesis methods can be divided into geometry-based and learning-based approaches. The former tries to first estimate (or fake) the approximate underlying 3D structures, followed by applying some transformation to the pixels in the input image to produce the output [1, 36]. The latter, on the other hand, argues that novel view synthesis is fundamentally a learning problem, because otherwise it is woefully under-constrained. More recently, neural radiance fields (NeRF) [18], which belong to the second category, have shown impressive performance on novel view synthesis of a specific scene by implicitly encoding volumetric density and color through a neural network. Starting from small-scales [18], scene-level NeRFs, such as Block-NeRF [28], are also proposed such that important use-cases, e.g., autonomous driving [2] and aerial surveying [12] are enabled by reconstructing large-scale environments. In contrast, our method takes the input as BEV semantics and text description and outputs multi-view perspective RGB.

## 3 OUR METHOD

Our method aims to generate multi-view perspective images from text prompts given pixel-level BEV semantic correspondences. Specifically, we denote the BEV semantics as $B \in \mathbb{R}^{H_b \times W_b \times c_b}$, with the ego car assumed to be located at the center. And $H_b$, $W_b$, and $c_b$

**Figure 2: MVPbev consists of two stages. The first stage projects BEV semantics to perspective view with camera parameters to maintain global semantic consistency. The second stage parses both perspective semantics and text prompts, and generates multi-view images with both visual consistency and test-time instance-level control by explicit enforcing in latent.**

are the height, width, and number of semantic classes of $B$, respectively. Our goal is to generate a set of perspective RGB images with resolution $H$ by $W$, or $\{I_m\}_m$ in particular, under $M$ virtual camera views. And the $m$-th perspective image is referred to as $I_m \in \mathbb{R}^{H \times W \times 3}$ where $m = \{1, \ldots, M\}$. In particular, we assume the intrinsics, extrinsic rotation, and translation of the $m$-th camera are given and denote them as $K_m$, $R_m$, and $T_m$, respectively.

As described above, we obtain visually coherent multi-view images by leveraging both global and local consistency in implicit and explicit manners. Specifically, our method consists of two stages. Our first stage takes BEV semantics $B$ as well as $\{K_m, R_m, T_m\}_m$ as input and projects BEV semantics to each perspective view w.r.t. its camera parameter set, denoting as $S_m \in \mathbb{R}^{H \times W \times c_b}$ for the $m$-th view. The second stage parses $\{S_m\}_{m=1}^M$ and text prompts as input. And it produces RGB images from $M$ perspective views. $I_m$ denotes the generated RGB image from $m$-th view. More specifically, our first projection stage enforces global semantic consistency between BEV and perspective view explicitly with the help of geometric transformation. Meanwhile, the generation stage imposes consistency implicitly among overlapping perspective views with a multi-view attention module. Finally, we propose to explicitly enforce the visual cues at overlapping FOVs to be coherent with our novel training initialization and de-noising designs. The overall pipeline of MVPbev can be found in Fig. 2. We provide more details of the first and second stages in Sec. 3.1 and Sec. 3.2 respectively. And the model training process is described in Sec. 3.3.

### 3.1 Semantic-consistent view projection

Assuming that diverse yet plausible BEV semantics $B$ can be obtained effortlessly with existing simulation methods [32], the first fundamental problem that our method should address is to maintain cross-view semantic consistency from $B$ to perspective images

$\{I_m\}_m$. Secondly, contents at overlapping FOVs should also be coherent. For example, not only the background classes, such as buildings or trees, but also the foreground road participants, should be of similar appearance when they appear in different views. To this end, we first propose to project BEV semantics to $M$ perspectives view with camera parameters, which generates $\{S_m\}_{m=1}^M$ perspective semantics. Compared to existing work [34], our projection step ensures semantic-wise consistency between BEV and perspective views with the help of geometry constraints, leading to fewer accumulative errors at the generation step. And our projection results can be found in Fig. 3.

### 3.2 View consistent image generation

Simply working on individual perspective semantics may lead to inconsistent content across different views, especially at overlapping FOVs. For instance, the buildings and the vegetation that appear at the FOVs among multiple views, e.g., the front, front-right, back, and back-left, have different appearances. This is due to the lack of interactions among cross-view cameras. We would like to note that such inconsistency would be reflected by neither BEV layout segmentation nor object detection metrics as it has influences on background classes only.

Motivated by this, we propose to focus on these overlapping areas both methodologically and experimentally. As for our method, we apply strong coherency constraints on the background content by estimating the homography of overlapping areas, followed by a multi-view attention module to implicitly enforce the styles at various views to be coherent w.r.t. estimated corresponding points. In this case, appearance consistency can be enforced not only on the background layout areas where the semantics are provided, but also on the other regions where control signals are missing. As for the evaluation purpose, we introduce human analysis to provide reliable evaluations on whether the generated images, especially the overlapping regions, are realistic or not. We demonstrate that

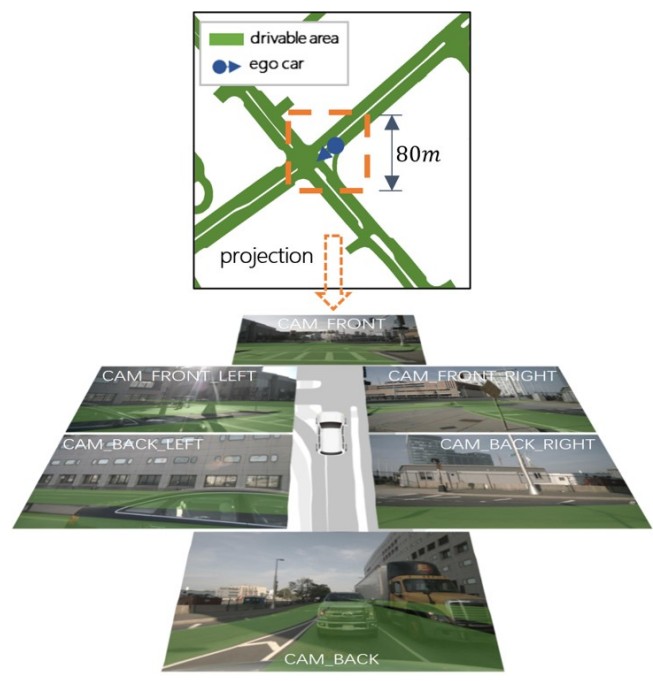

**Figure 3: We visualize our BEV project process. Given a BEV semantic map $B$, we project it to multiple perspective views. We overlay the semantics on the original RGB images in perspective view for better comparison.**

our proposed method copes with background consistency well (See Sec.4 for quantitative and qualitative results).

*Homography estimation.* We take the initial step towards enforcing visual consistency at overlapping FOVs by estimating the overlapping regions. To this end, we propose to compute the homography between images with overlapping FOVs. As illustrated in many driving datasets, one view generally overlaps with views on its left and right sides. Therefore, for the $m$-th view, we only need to consider $m_l = \text{mod } (m+M-2, M)+1$ and $m_r = \text{mod } (m+1, M)$, which are the left and right views of the $m$-th view respectively. Then we estimate the homography from view $m_r$ to view $m$ and denote the mapping function as $H_m$. Consequently, the $p = [x, y]$ coordinate in the $m$-th view will be mapped to coordinate $\hat{p} = [\hat{x}, \hat{y}]$ in view $m_r$. Or $\hat{p} = H_m(p)$. Similarly, we define an inverse mapping $\overline{H}_m$ that maps $\hat{p}$ in $I_{m_r}$ to $p$ in $I_m$.

*Multi-view attention module.* What makes a set of views unrealistic? The first and foremost thing is the incoherence among images. In other words, realistic ones must appear consistent, as if they were taken in the same physical location at the same time of a day. More specifically, the visual styling of this set of images needs to be consistent such that all of them appear to be created in the same geographical area (e.g., urban vs. rural), time of day, with the same weather conditions, and so on. To this end, we introduce a multi-view attention module such that when generating the RGB from the $m$-th view, the views on its left and right sides are considered. For a token located at position $p$ in the feature map $F_m$ generated

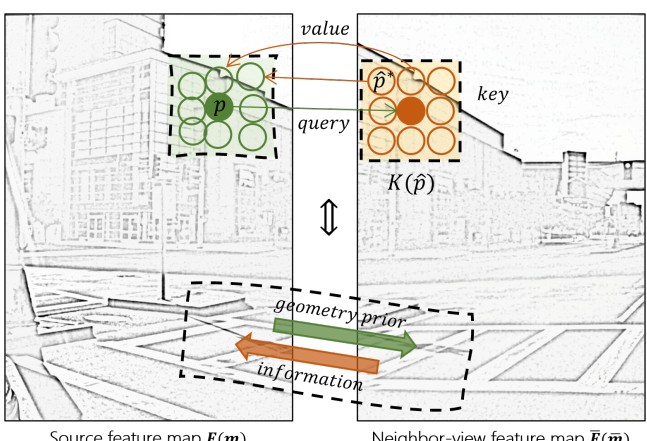

Source feature map $F(m)$      Neighbor-view feature map $\overline{F}(\bar{m})$

**Figure 4: Our multi-view attention module implicitly exploits the cross-view consistency. Specifically, it aggregates information from the target feature pixels in neighbour views, which are obtained by homographic transformation, to the source.**

from $m$-th view, we compute the attention output based on the corresponding pixels $K(\hat{p})$ in the feature maps generated by view $\bar{m} \in \{m_r, m_l\}$, where $\hat{p}^* \in K(\hat{p})$ denotes a $K$ by $K$ region centered at $\hat{p}$.

Mathematically, we follow a similar formulation as in [30] and define our multi-view attention module as:

$$\mathbf{a} = \sum_{\bar{m}} \sum_{\hat{p}^*} Softmax([QF_m(p) \cdot [K\bar{F}_{\bar{m}}(\hat{p}^*)])V\bar{F}_{\bar{m}}(\hat{p}^*), \quad (1)$$

where $Q$, $K$, and $V$ are the learnable weights of query, key, and value matrices respectively. $\bar{F}_{\bar{m}}(\hat{p}^*) = F_{\bar{m}}(\hat{p}^*) + d(\overline{H}_m(\hat{p}^*) - p)$. We further denote $d(\cdot)$ as a position encoding to the $F_{\bar{m}}(\hat{p}^*)$ based on the 2D displacement between $p$ and $\overline{H}_m(\hat{p}^*)$. As can be found in Eq. 1, our multi-view attention module aims to aggregate information from the target feature pixels $K(\hat{p})$ to $p$. We provide a simple illustration of our multi-view attention module in Fig. 4.

### 3.3 Model training and inference

To train our model, we introduce a multi-view Latent Diffusion Models (LDMs) [24] loss. Basically, the original LDMs consist of a variational autoencoder (VAE) with encoder $\mathcal{E}$ and decoder $\mathcal{D}$, a denoising network $\delta_\theta$ and a condition encoder $\tau_\theta$. The input image $I_m$ is mapped to a latent space by $\mathbf{l}_m = \varepsilon(I_m)$, where $\mathbf{l}_m \in \mathbb{R}^{h \times w \times c}$. We follow the routine to set $\frac{H}{h} = \frac{W}{w}$ and they both equal to 8. Later on, the latents will be converted back to the image space by $\tilde{I}_m = \mathcal{D}(\mathbf{l}_m)$. The denoising network $\delta_\theta$ is a time-conditional UNet, which leverages cross-attention mechanisms to incorporate the condition encoding $\tau_\theta(\mathbf{c})$. In our case, $\mathbf{c}$ consists of text-prompt and semantics in perspective view $S_m$.

For each training step, we firstly uniformly sample a shared noise level $t$ from 1 to $T$ for all the multi-view images $\{I_m\}_{m=1}^M$, denoting them as $\{\epsilon_m^t\}_m$. And $\epsilon_m^t \sim \mathcal{N}(0, 1)$. To leverage the cross-view consistency, we further enforce these noises to be the same if they correspond to the same pixel. Starting from the first view, or $m = 1$,

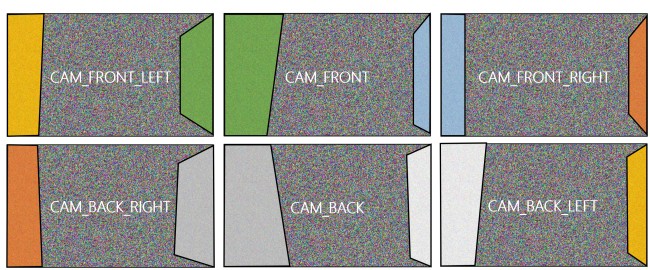

**Figure 5: We explicitly enforce that the noise value of pixels at overlapping FOV should be consistent across views.**

we re-assign value at coordinate $x, y$ of $\epsilon_m^t$, or $\epsilon_m^t(x, y)$, to $\epsilon_{m_r}^t(\hat{x}, \hat{y})$. And we repeat the process until $m_r = 1$. We provide one example set of our initialized $\{\epsilon_m^t\}_{m=1}^M$ in Fig. 5. Finally, our model training objective is defined as:

$$\mathcal{L} := \mathbb{E}_{\{1_m, \epsilon_m^t\}_{m=1}^M, t, \mathbf{c}} \left[ \sum_{m=1}^M \|\epsilon_m^t - \delta_\theta^m(\{l_m^t\}, t, \tau_\theta(\mathbf{c}))\|_2^2 \right], \quad (2)$$

where $\delta_\theta^m$ is the estimated noise for the $m$-th image. And we use $l_m^t$ to denote the noisy latent for the $m$-th image.

At sampling time, the denoising (reverse) process generates samples in the latent space, and the decoder $\mathcal{D}$ produces RGB images with a single forward pass. To incorporate our idea that pixels at overlapping regions should be visually similar even at different views, we again employ the value assignment process. Similar to the noise initialization step, we re-assign value at coordinate $x, y$ of $l_m^t$, or $l_m^t(x, y)$, to $l_{m_r}^t(\hat{x}, \hat{y})$. This re-assignment starts from $m = 1$ and does not stop until $m_r$ equals to 1. In experiments, we observe that our design would improve the visual results if applied to up to $\frac{3 \times T}{5}$ denoising steps. Otherwise, the performance deteriorates.

**Inference** As claimed in above, MVPbev can be extended with instance-level controllability. Specifically, Our MVPbev allows the user to click on the target instance and provide color-specific requirements. To achieve this, we propose a special mechanism for multiple foreground objects control, which manipulates the response of cross-attention layers to accurately guide instance-level synthesis. Assuming that instance-level masks can be obtained at each view with either existing methods [5] or simple retrieval. Specifically, we first obtain the instance-level and scene-level latents individually with its paired prompt. Then they are effectively combined with those binary instance-level mask, leading to more spatial-consistent performance. Please note that such ability on foreground objects of MVPbev are training-free, leading to far better extendability and test-time controllability. We refer the readers to supplementary for more details.

## 4 EXPERIMENT

### 4.1 Experiment setup

**Dataset** We validate our ideas on NuScenes dataset [2] where full 360-degree coverage provided by six cameras is available. Specifically, it consists of 1000 examples of street-view scenes in Boston and Singapore, each of which lasts for 20s and is captured at 12Hz. Besides 1.4M camera images, NuScenes also provides multi-modal

data, including both global map layers and 3D object bounding boxes annotated on 40k keyframes. We follow the standard split of 700/150/150 for training, validation, and testing. We report our results on the validation set of Nuscenes and follow the split of [2] where 600 sets of images are used.

**Evaluation metrics** We follow the design of [29] to include image quality of generated images as well as their visual consistency in our evaluation metrics. In addition, semantic consistency is also valued in our metrics as it reflects the synthesis quality of different semantic categories.

- *Image quality* is measured by Fréchet Inception Distance (FID) [6], Inception Score (IS) [25], and CLIP Score (CS) [22]. In particular, FID is based on the distribution similarity between the features of the generated and real images. The IS measures the diversity and predictability of generated images. Finally, CS measures the text-image similarity according to pre-trained CLIP models [22].
- *Visual consistency* provides pixel-wise similarity measurements on overlapping regions. We borrow the idea from Peak Signal-to-Noise Ratio (PSNR) where we first compute this PSNR between all the overlapping regions, and then compare this "overlapping PSNR" for ground truth images and generated images. The higher this value is, the better visual consistency will be. Note that the process of computing "overlapping PSNR" is based on estimated homography matrices, it's possible that generated image yields higher values than ground truth image.
- *Semantic consistency* measures the pixel-wise semantic consistency between generated images and ground truth. In our case, we utilize the Intersection-over-Union (IoU) score. Particularly, we report the semantic IoU both in perspective view and BEV. As for the former, we apply pre-trained segmentation model [5] on generated images, leading to semantic predictions in perspective view. These predictions are then compared with $\{S_m\}_m$ to obtain IoU in perspective view. As for the latter, we apply pre-trained CVT [35] to generated images and the BEV IoU is obtained by comparing predictions from CVT with $B$.
- *Object-level controllability* measures how accurately the object-instance is generated w.r.t. test-time descriptions. Here we report the averaged color distance Delta-E in CIELAB color space as well as their standard deviations.

Besides these metrics, we also perform human analysis. We request humans to decide which method is more visually realistic and consistent when results from different methods are provided. Please note that method is anonymous to humans and when compared, we ensure that the same input control signals are provided to various methods. Meanwhile, we also conduct experiment with instance-level controllability. Humans are provided with objects as well as their targeted color, paired with the generated images. And they will vote whether the generated objects meet the requirements.

**Baselines** We select the following two state-of-the-art methods as our baselines for thorough comparisons:

- *SD+ControlNet* [24, 34] is a basic yet powerful image generation model. In our experiment, we work on the projected $\{S_m\}_m$ to avoid domain gaps from different viewpoints. Starting from a pre-trained ControlNet [34], this baseline is fine-tuned on NuScenes training set.
- *MVDiffusion* [29] is proposed to generate multi-view consistent images and achieves good performance on tasks like panoramic

**Table 1: Quantitative results and human analysis on NuScenes. We observe a noticeable superiority of our MVPbev.**

(a) Quantitative results on NuScenes dataset

| Method | Training Samples | Image Quality | | | Semantics Consistency | Visual Consistency |
|---|---|---|---|---|---|---|
| | | FID↓ | IS↑ | CS↑ | IoU$_{BEV}$↑ | PSNR↑ |
| Reference-score | - | 6.20 | 5.77 | 27.54 | 0.711 | 15.37 |
| BEVGen[27] | 28130 | 25.54 | - | - | 0.502 | - |
| MagicDrive[9] | 28130 | **16.20** | - | - | **0.611** | - |
| Controlnet [34] | 6000 | 21.93 | 4.71 | 27.02 | 0.434 | 12.82 |
| MVD [29] | 6000 | 19.89 | 4.91 | 27.07 | 0.440 | 12.66 |
| Ours | 6000 | 16.95 | **6.35** | **28.79** | 0.510 | **20.67** |

(b) Human analysis on NuScenes dataset.

| Comparisons | | Win | Undecided | Lose |
|---|---|---|---|---|
| Cross-view Consistency | MVD v.s. Controlnet | .26 | .50 | .25 |
| | Ours v.s. Controlnet | .73 | .27 | .00 |
| | Ours v.s. MVD | .71 | .29 | .00 |
| Camera Pose Consistency | Ours v.s. MagicDrive | .61 | .31 | .08 |

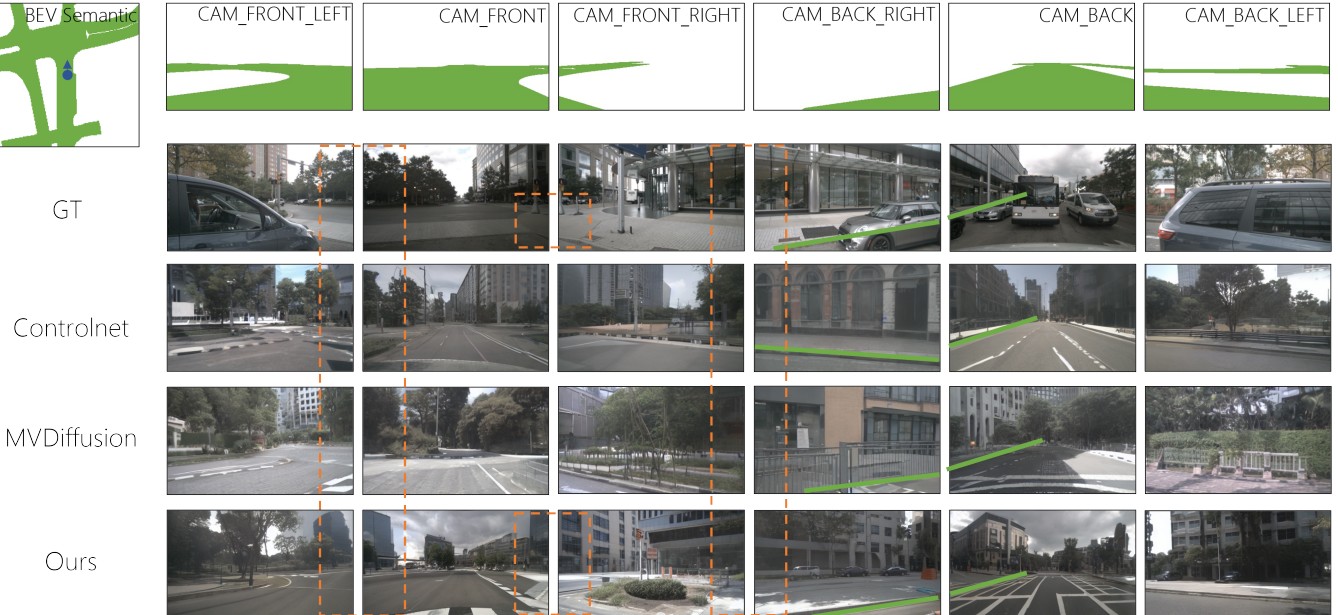

**Figure 6: Our MVPbev is able to achieve the most visual and semantic consistent images w.r.t. input control signal. We highlight the overlapping area and road boundary in orange bounding boxes and green lines.**

image generation. However, it is designed neither for dramatic view changes, e.g., BEV to perspective view, nor for semantic control signals. To this end, we follow our pipeline to first map BEV semantics to perspective view and then update MVDiffusion with a pre-trained ControlNet [34] backbone. Specifically, We re-implement [29] based on its official code and fine-tune it on NuScenes training images.

- *BEVGen* [27] is the first step towards road scene generation, where the control signals are limited to BEV semantics and camera parameters. The full dataset is used for training.
- *MagicDrive* [9]is the most recent published work on road scene generation. We use their released model for efficient comparison. Please note that we use only 20% images uniformly sampled from dataset for training while they use the full dataset.

### 4.2 Implementation Details

Our BEV semantics $B$ reflects an 80m × 80m space with ego car located at the center position. $c_b$ represents the drivable area in NuScenes. The resolution of the perspective image is $H \times W = 256 \times 448$, leading to $h \times w = 32 \times 56$. As for the hyper-parameters, we set

$M$ and $K$ to 6 and 3 respectively. We have implemented the system with PyTorch [21] while using publicly available Stable Diffusion codes [31]. Specifically, it consists of a denoising UNet to execute the denoising process within a compressed latent space and a VAE to connect the image and latent spaces. The pre-trained VAE of the Stable Diffusion is maintained with official weights and is used to encode images during the training phase and decode the latent codes into images during the inference phase. In experiments, we use a machine with 1 NVIDIA A40 GPU for training and inference. Batch size is set to 6 and $T$ equals to 50. We refer the readers to supplementary for full implementation details.

### 4.3 Multi-view BEV generation

We compare our MVPbev with baselines and report the performance in Table 1. The first row in this table is obtained on ground truth images. For instance, we split the ground truth validation images into two halves, and then obtain the FID score by taking one split as ground truths and the other half as generated images. As for the IoU scores, we apply the Mask2Former [5] and CVT [35] on validation images and compare their predictions with ground truths

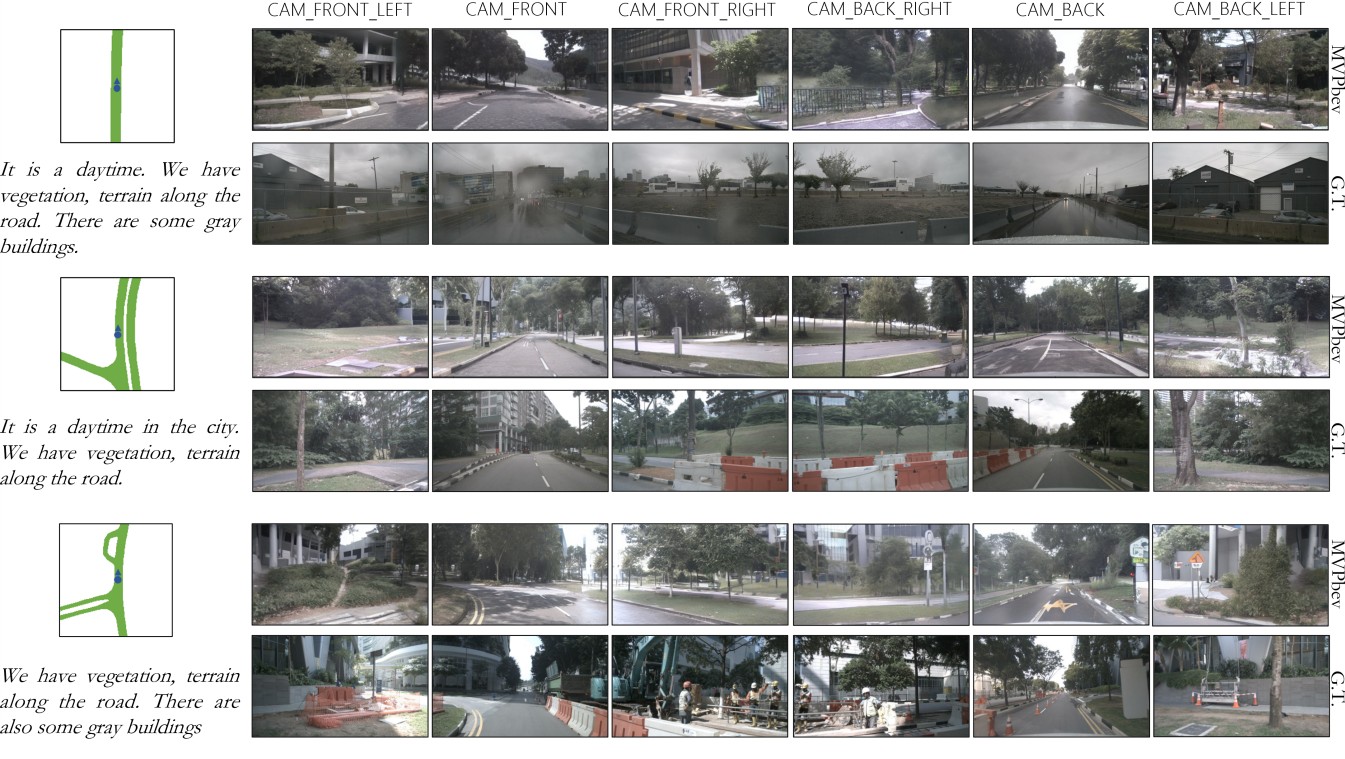

**Figure 7: Qualitative results of MVPbev. We provide three sets of examples. The leftmost column consists of input signals, while the 2nd to 7th columns are images from different perspective views.**

$\{S_m\}_m$ and **B**. As can be found in this table, our MVPbev almost always ranks first among comparable baselines, such as Controlnet and MVD. And we even achieve comparable results w.r.t. SOTA methods that are trained with far more training data.

We also provide visual comparisons to existing baselines in Fig. 6. The top row showcases the bev semantics $B$ as well as the projected semantics $\{S_m\}_{m=1}^{M}$ in perspective views. We also provide the ground truths as well as multi-view images generated by other methods from the second to the fifth row. Our method MVPbev, compared to other baselines, produces the most consistent perspective images, especially at overlapping FOVs. As highlighted by orange bounding boxes and green lines, our MVPbev not only

generates perspective images that are consistent with semantic guidance, but also maintains high visual consistency across multiple views. Such consistency is more visible and valuable for pixels that appear in different views.

**Qualitative results** Besides quantitative results, we also provide qualitative examples in Fig. 7. As can be found in this figure, our MVPbev is able to generate visually consistent images from diverse bev semantics and text prompts. Compared to ground truth, ours can obtain satisfactory consistency at overlapping FOVs. We refer the readers to supplementary for more visual examples about the controllability over BEV and text prompt.

*Test-time controllability and generalizability.* **View-point generalizability** As described before, one of the main drawbacks of the existing work is the lack of ability in terms of handling view-point changes at test time. To showcase our ability, we revise the camera extrinsic during inference and check whether the results would change accordingly. In practice, we rotate the all $M$ camera by $\{-25°, -15°, -5°, 5°, 15°, 25°\}$ w.r.t. the head direction of the ego car. mimicking potential different setups of camera mounting. This is equivalent to changing the $\{R_m\}_m$ in our input signal. We randomly generate 200 sets of images for each rotation angle and provided the generated results from MagicDrive [9] and ours to humans. Qualitative results are provided in Fig. 9. We overlay the projected semantics in each view for better visualization. Not surprisingly, prior art merely follows the control signal. While MVPbev

**Figure 8: MVPbev is able to handle test-time instance-level controllability with various color requests.**

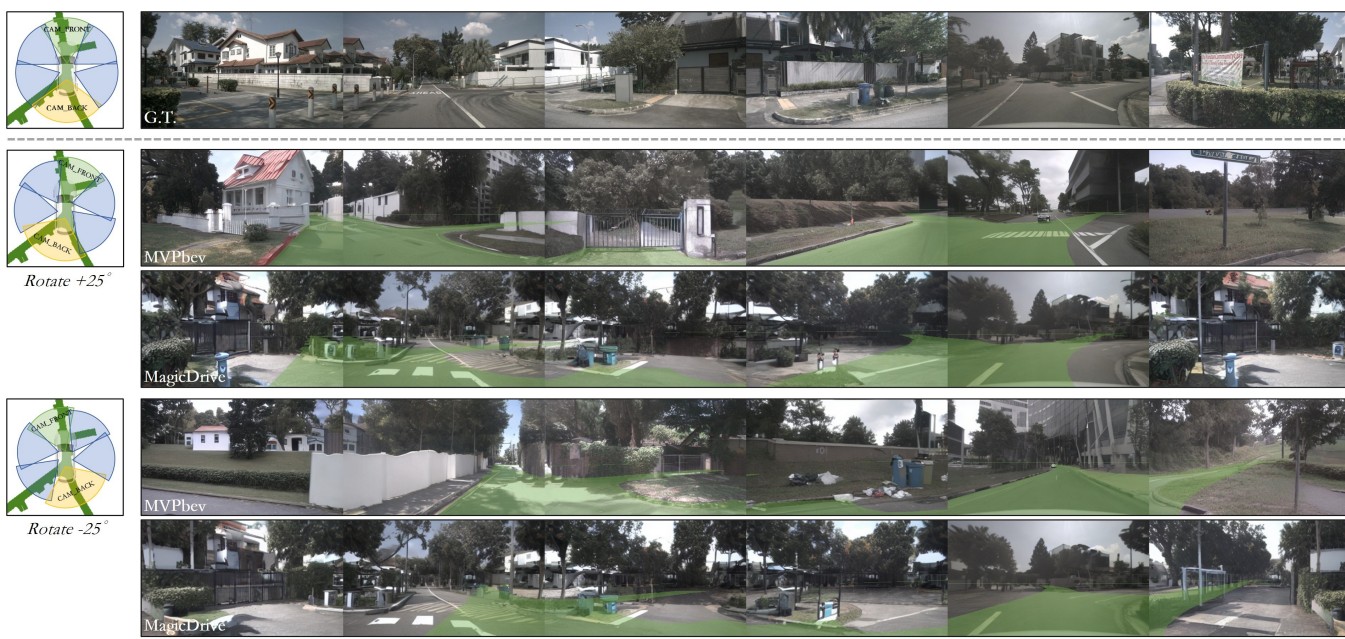

**Figure 9: MVPbev is robust to camera pose changes without re-train the entire model, providing better test-time generalizability.**

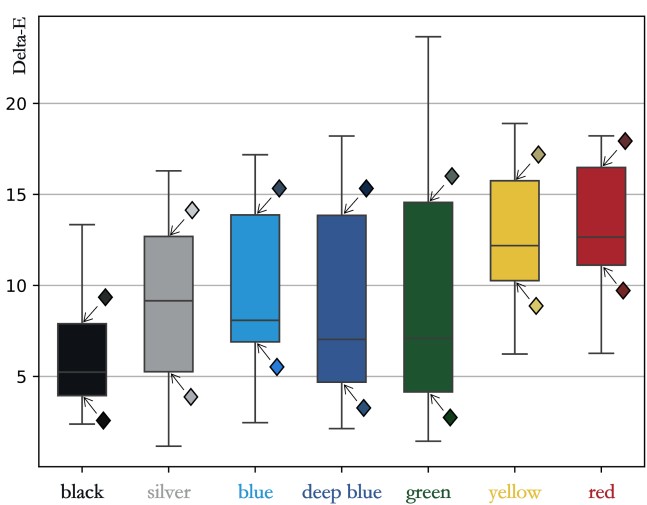

**Figure 10: The Delta-E distance between the ground truth and generated color on our controlled instances.**

gives superior results considering the semantics, demonstrating better test-time generalizability. And this observation is also supported by our detailed human analysis in Table 1.

**Object-level controllability** A practical generative model should be a controllable one. To this end, we conduct another experiment to showcase the object-level controllability. In this experiment, we include an additional description of the object color in the original text prompt and then check whether such control can be reflected in generated scenes at test time. In experiment, we randomly choose 151 set of images, including 195 object instances, and provide random color requests out of seven popular colors for vehicles. We

report our qualitative evaluations in Fig. 10 and qualitative examples in Fig. 8 respectively. Though the Delta-E seems to be noticeable, we argue that this is mainly due to the de-noising process where colors of vehicles are in harmony with the environment, e.g., less bright in rainy days. This is supported by our visual results as well as human analysis.

*Human analysis.* Compared to evaluation metrics, human analysis provides a more reliable tool for image quality measurement. Therefore, we conduct a comprehensive human analysis of our tasks. Specifically, we provide two sets of generated images, which are generated from two different methods with the same input signal, to humans. Then we ask them to decide which set of images is better, considering the image quality and visual consistency. As can be found in Table 1, our MVPbev outperforms baselines significantly, indicating that we can indeed generate photo-realistic yet consistent images. Meanwhile, we report the test-time viewpoint changes by comparing ours to MagicDrive [9], showing that MVPbev provides better generalizability quantitatively. Finally, we ask humans to decide whether the generated instance color can be regarded as the requested one. In our experiment, 93.5% of the instances are voted as correctly generated. We refer the readers to supplementary for more details about human analysis.

## 5 CONCLUSION

Our goal is to generate multi-view perspective RGB from text prompts given BEV semantics. To this end, we introduce a two-stage method MVPbev to first project BEV semantics to perspective views and then perform image generation w.r.t. both text prompts and individual perspective semantics. Specifically, we propose a novel initialization and denoising processes to explicitly enforce local consistency at overlapping FOVs. Results showcase the superiority of MVPbev under various metrics and test-time generalizability.

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
