# OpenReview forum: "MVPbev: Multi-view Perspective Image Generation from BEV with Test-time Controllability and Generalizability"
_acmmm.org/ACMMM/2024/Conference — MM2024 Poster_

### Official Review · Reviewer_wUzG · 2024-05-14

**Rating:** 5
**Confidence:** 4

**Summary:**

The paper proposes a novel two-stage method MVPbev that aims to generate controllable multi-view perspective RGB images with given BEV semantics and text prompts by explicitly enforcing cross-view consistency. Specifically, it consists of view projection and scene generation stages. For the view projection stage, it projects BEV semantics to perspective view with camera parameters to maintain global semantic consistency. For the scene generation stage, it parses both perspective semantics and text prompts, and generates multi-view images with both visual consistency and test-time instance-level control by explicit enforcing in latent. Through extensive experiments on NuScenes, the MVPbev method demonstrates better test-time controllability and generalizability, surpassing the state-of-the-art methods under various evaluation metrics.

**Strengths:**

1. Overall: The paper proposes a novel multi-view image generation method MVPbev to produce semantically and visually consistent perspective RGB images from BEV semantics, with only thousands of images as training data. The controllability and generalizability of the MVPbev method are evaluated through extensive experiments through extensive experiments on NuScenes, surpassing the state-of-the-art methods under various evaluation metrics.
2. Better Test-time Generalizability: As for conditional image generation, existing works can’t achieve test-time generalizability, e.g., view-point changes or detailed instance-level text prompts. In contrast, the proposed method exploits both global and local consistency, together with a training-free objects control method to enforce detailed instance-level control.
3. Robust Evaluation: Existing works of conditional image generation don’t conduct human analysis on image generation quality. In contrast, the paper provides comprehensive human analysis to demonstrate the effectiveness of the proposed method in a more reliable manner.

**Limitations:**

1. Insufficient Analysis of Experimental Results: Specifically, there is no thorough examination of cases where the proposed method MVPbev failed to achieve state-of-the-art results when compared against methods like BEVGen and MagicDrive. A detailed exploration of the factors leading to these discrepancies is essential to fully understand the limitations and identify potential areas for improvement in the proposed method.
2. Need for Analysis of Imperfect Sample Cases: While the paper includes visual comparisons with existing baselines in Figure 6 and qualitative examples in Figure 7, it only mentions that the MVPbev produces the most consistent perspective images, particularly at overlapping fields of view (FOVs). However, I have observed noticeable inconsistencies between some images generated by the proposed method and the ground truth, which are not adequately analyzed by the authors. Incorporating such an analysis would be beneficial in demonstrating the method's limitations and suggesting areas for improvement.
3. Lack of Ablation Studies: The paper lacks a dedicated section or discussion on ablation studies, which makes it challenging to assess the effectiveness of various components like the multi-view attention module, special initialization, training-free objects control method, and the denoising processes. Without individual validation of these elements, it is difficult to determine the contribution of each to the overall performance of MVPbev.
4. Language and Grammar Errors:
Line 103: “the remaining question is to generate cross-view…” -> “the remaining question is how to generate cross-view…”
Line 331: “lack of interactions” -> “lack of interaction”

**Suitability:**

3

---

### Official Review · Reviewer_etDR · 2024-05-21

**Rating:** 4
**Confidence:** 2

**Summary:**

The paper proposes a novel multi-view image generation method (MVPbev) to address the multi-view perspective RGB generation from text prompts given Bird-Eye-View(BEV) semantics.

**Strengths:**

MVPbev projects the given BEV semantics to perspective view with camera parameters, empowering the model to generalize to unseen viewpoints.  MVPbev introduces the multi-view attention module where special initialization and denoising processes are introduced to explicitly enforce local consistency among overlapping views w.r.t. cross-view tomography. The method further allows test-time instance-level controllability by refining a pre-trained text-to-image diffusion model. The method obtians state-of-the-art performances on large driving datasets.

**Limitations:**

(1) The analysis of ablation experiments is not exhaustive.
(2) Some parameter symbols are not clearly explained：parameter c of Eq(2).
(3)The experimental results need to explain in detail: The author did not thoroughly analyze the experimental results of table 1, nor did they explain the advantages of their method. In Fig 10, the author also did not explain and describe the content.
(4) Some comparative methods （MVD, Controlnet）have not been tested on this specific dataset; the author should provide the settings for these methods on this dataset, including parameters, experimental environment, etc.

**Suitability:**

2

---

### Official Review · Reviewer_JSaN · 2024-05-26

**Rating:** 5
**Confidence:** 4

**Summary:**

The paper is about a method named MVPbev, which stands for Multi-view Perspective Image Generation from Bird's-Eye-View (BEV) with Test-time Controllability and Generalizability. The MVPbev method introduces a two-stage design. The first stage projects the given BEV semantics into perspective views using camera parameters. The second stage involves a multi-view attention module with special initialization and denoising processes to enforce local consistency among overlapping views with respect to cross-view homography.

**Strengths:**

1 The introduction of specific initialization and denoising processes to enforce local consistency at overlapping fields of view (FOVs) is an innovative solution to a common challenge in multi-view image generation. By using a multi-view attention module and enforcing consistency through geometric transformations and homography estimation, the method ensures that overlapping regions in different views are visually coherent.

2 Test-Time Instance-Level Controllability: MVPbev enables changes in viewpoint and instance-level detailed text prompts at test time without the need for additional training data, providing greater flexibility and adaptability.

**Limitations:**

1 It lacks comparative experiments with Controlnet and MVD in terms of controllability.
2 Since the title includes the Test-time Controllability, then how to do it should be placed in the main text. Can this mechanism only control colors effectively? It seems that this mechanism can also be applied to ControlNet.
3 It seems that there is no specific mechanism to achieve Test-time generalizability.

**Suitability:**

3

---

### Official Review · Reviewer_iFQi · 2024-05-30

**Rating:** 4
**Confidence:** 2

**Summary:**

This paper introduces a method called MVPbev, which generates consistent multi-view perspective images from bird's-eye view (BEV) semantics and text prompts through a two-stage design. This method addresses the shortcomings of existing methods in terms of viewpoint variation and detailed control. Its superior generation performance and control during testing have been validated through multiple evaluation metrics and manual analysis on the NuScenes dataset.

**Strengths:**

1. The novelty is sufficient, focusing on generating multi-view images for real-time autonomous driving.
2. The writing and diagrams are easy to understand.
3. The main experiments are detailed and compare many existing algorithms for autonomous driving scene generation.

**Limitations:**

Lacks an ablation study for modules such as the Multi-view attention module and Homography estimation.

**Suitability:**

2

---

### Meta-Review · Area_Chair_XMEv · 2024-06-30

**Recommendation:** Accept (Poster)
**Confidence:** 5

**Metareview:**

The paper receives two borderline accept and two weak accept. It is a clear acceptance case.